# Effects of geomagnetic mirror force and pitch angles of precipitating electrons on ionization of the polar upper atmosphere

Tomotaka M. Tanaka<sup>1,2</sup>, Yasunobu Ogawa<sup>1,2</sup>, Yuto Katoh<sup>3</sup>, Mizuki Fukizawa<sup>2</sup>, Anton Artemyev<sup>4</sup>, Vassilis Angelopoulos<sup>4</sup>, Xiao-Jia Zhang<sup>5</sup>, Yoshimasa Tanaka<sup>1,2,6</sup>, Akira Kadokura<sup>2,6</sup>

<sup>5</sup> The Graduate University for Advanced Studies, SOKENDAI, Japan

<sup>2</sup>National Institute of Polar Research, NIPR, Japan

<sup>3</sup>Tohoku University, Japan

<sup>4</sup>University of California, Los Angeles, UCLA, United States of America

<sup>5</sup>University of Texas, Dallas, UTD, United States of America

<sup>6</sup>Joint Support-Center for Data Science Research, ROIS, JAPAN

Correspondence to: Tomotaka Tanaka (tanaka.tomotaka@nipr.ac.jp)

Abstract. We studied the effects of the geomagnetic mirror force on electron density enhancements in the polar atmosphere due to energetic electron precipitation. Using the pitch angle and energy distribution of electrons observed by the low-altitude Electron Losses and Fields INvestigation (ELFIN) satellites as initial conditions, the electron density in the atmosphere caused by precipitating electrons was calculated using a simulation with two different methods: a traditional method that does not include the effect of the mirror force and a recently developed method that includes the effect. From a simultaneous observation event of the ELFIN satellite and the European Incoherent SCATter scientific radar system (EISCAT) Tromsø radar, it was found that the method with the effect of the mirror force reduces electron density by about 40% at an altitude of 80 km compared to the traditional method. This decrease was pronounced when the pitch angle distribution of high-energy electrons was concentrated in the pitch angle range of the trapped component and near the loss cone. The maximum decrease was 50%. For an event where the altitude profile of electron density was accurately determined from the EISCAT radar, the electron density profile estimated using the method with the effect of mirror force showed better agreement with the electron density profile derived from the EISCAT radar. The comparison between simulation results and the observation data contributes to the establishment and improvement of atmospheric ionization models using various types of precipitating electrons.

#### 1 Introduction

25

The energetic (> 50 keV) electron precipitation (EEP) has attracted attention due to its impact on mesospheric ionization and ozone chemistry. Miyoshi et al. (2020) suggested that precipitating energetic electrons penetrate to lower altitudes with the appearance of diffuse auroras. Oyama et al. (2017) reported electron density enhancements in the lower D-region ionosphere during pulsating auroral events, attributed to EEP. These enhancements may accelerate chemical reactions and the production of nitrogen or hydrogen oxides, which destroy ozone in the mesosphere below 80 km altitude (Turunen et al., 2016). Thus, it

has been suggested that EEP has ability to make chemical changes in the lower altitude based on both observations and simulation (Miyoshi et al., 2021; Ozaki et al., 2022; Murase et al. 2022). However, modeled ionization rates due to EEP can differ by up to an order (Nesse et al., 2022). Therefore, it is necessary to evaluate the effects of EEP on the atmosphere accurately by considering simulation improvements.

The mirror force, one of the effects of the magnetic field on charged particles, has recently been considered in numerical simulations of atmospheric ionization by precipitating electrons. As the magnetic field strength at atmospheric altitudes is nearly constant, the change in the pitch angle of precipitating electrons by the effect of the mirror force has been considered negligible (Rees, 1963). Even if some numerical simulations included the pitch angle, it was only used to calculate the altitude change of the electron motion using the inclination of the magnetic field and pitch angle. Thus, velocity changes due to the mirror force were still ignored (Solomon, 1993, 2001). A numerical simulation to calculate the altitude profile of atmospheric ionization rate by EEP, including the effect of the mirror force, was developed (Lehtinen et al., 1999), and the results of the simulation were parameterized by the pitch angle and energy of the electron (Xu et al., 2020). Furthermore, using a newly developed simulation, Katoh et al. (2023) showed the concrete difference in ionization rates between cases with and without the effect of the mirror force. By comparing the simulation results with and without the effect, it was found that when an electron with a large pitch angle had energies above 100 keV, the ionization rate of the former was less than 10% of the latter (Katoh et al., 2023). However, this effect has not yet been verified or confirmed by any observational data.

Although some satellites have observed electrons' energy and pitch angle distribution, it has been challenging to use them in simulation as input. The Arase satellite, for example, is too far away to observe the pitch angle distribution of electrons reaching the earth's atmosphere in detail, while the energy range of electrons observed by the Reimei satellite was too low to make differences between cases with and without the effect of the mirror force, as shown in Katoh et al. (2023). However, the ELFIN (Electron Losses and Fields INvestigation CubeSats) satellites, polar-orbiting and low-altitude satellites, observed both pitch angle and energy distribution of energetic electrons. Therefore, it has been possible to evaluate the actual atmospheric effects by combining the satellites with the simulations developed by Katoh et al. (2023).

This study aims to evaluate the effect of the mirror force on atmospheric electron density integrating satellite observations with simulation results and subsequently validating the results with radar data. Even though it is challenging to evaluate the ionization rate, calculating the electron density from the ionization rate enabled us to compare it with observational data. Using the pitch angle and energy distributions of electrons observed by the ELFIN satellites, we calculated the electron density enhancement due to the precipitating electrons with and without the effect separately to compare them with the actual data observed by the EISCAT (European Incoherent Scatter) Tromsø radar. In this study we mainly show comparison results with simulations focusing on the event of December 16, 2021, when the altitude profile of electron density was obtained accurately from EISCAT observations.

#### 2 Methods and Instruments

#### 65 2.1 ELFIN satellites

We used data from the ELFIN satellites. The satellites were launched in September 2018 and completed observations in September 2022. They were on a low-altitude (~450 km altitude) polar orbit (~93°inclination), and the orbital period is about 90 min. It consists of two CubeSats (ELFIN-A/B), flying in nearly identical orbits with a time difference of less than 20 min (Angelopoulos et al., 2020). Each satellite had an energetic particle detector for electrons (EPDE), which measured 50 - 7000 keV electrons with  $\Delta E/E < 40\%$  energy resolution and  $22.5^{\circ}$  pitch angle resolution. Since the electron flux above 2500 keV is negligible compared to electrons with energies below that, we used the electron flux data between 50 and 2500 keV in this study. The spin axis is kept perpendicular to the orbital plane, and the whole pitch angle is observed twice every spin period (~2.85 s) (Zhang et al., 2022). A loss cone angle  $\theta_{LA}$  is defined as a equation

$$\theta_{L.A.} = \sin^{-1} \sqrt{\frac{B}{B_{100}}} \tag{1}$$

where B, and  $B_{100}$  are the magnetic field strength at the satellite location and at 100 km altitude, respectively. In other words, it is the maximum pitch angle of electrons at the satellite point which can enter the atmosphere below 100 km altitude while moving in the magnetic field. The IGRF model (Alken et al., 2021) was used as a magnetic field model to calculate the loss cone angle.

#### 2.2 EISCAT Tromsø radar

We also used the EISCAT Tromsø UHF radar, which has been in operation since 1981 at Tromsø (69.6°N, 19.2°E, and magnetic latitude of 66.2°) in northern Scandinavia. This radar observes the altitude profile of electron density and other ionospheric parameters (Folkestad, K., 1983). For all events in this study, the beata pulse code was utilized, and the beam was directed along to the magnetic field line. The electron density data used in this study were obtained with the beam aligned along the magnetic field lines, and were derived from the power profile data using the standard analysis software GUISDAP (Lehtinen and Huuskonen, 1996). Appropriate power calibration was performed by comparing the radar-derived electron densities with the co-located ionosonde measurements (foF2 and foE), and calibration factors were applied accordingly.

#### 2.3 Simulation

The simulation used in this study is a Monte Carlo simulation developed by Katoh et al. (2023). It has atmospheric data of oxygen atoms, oxygen molecules, and nitrogen molecules using the NRLMSIS 2.1 emprical mode (Lucas, 2022; Emmert et al., 2022). It enables us to calculate the altitude profile of the ionization rate above Tromsø (L=6.45), using precipitating electron parameters—altitude, pitch angle, and energy—as initial conditions. The effect of the Lorentz force can be included or excluded manually when calculating particle transport. When considering the force, the magnetic field at a distance of Larmor

radius from the magnetic field line leading to Tromsø was given to satisfy  $\nabla \cdot \mathbf{B} = 0$  and used in the calculations (see Katoh et al., 2023).

#### 95 2.4 Combination of the ELFIN satellites data and the simulation






First, we searched for events in which either of the ELFIN satellites approached the EISCAT Tromsø radar. We used the IGRF model to calculate the geomagnetic footprints of the satellite, and selected events in which the geomagnetic latitude was within  $69.6^{\circ}$ N  $\pm$  2°, and the longitude was within  $19.2^{\circ}$ E  $\pm$  5°. Second, we selected events that showed some electron density peak under 100 km altitude or had a density of more than  $10^{10} \text{ m}^{-3}$  at 85 km altitude in the EISCAT data, which should be caused by the EEP. Next, we used the differential number flux of electrons in each energy and pitch angle bin observed by the satellite as the initial condition of the simulations. For each bin, we simulated the altitude profile of collision rate R [m<sup>-1</sup>], defined as the number of ionization events produced by a single precipitating electron in each 1 m altitude interval in the atmosphere. For some specific bins containing the loss cone angle, the collision rate was calculated more precisely by dividing the pitch angle range of the bin into ten parts and averaging the results. Katoh et al. (2023) described the method for calculating the collision rate and demonstrated that it varies sensitively with pitch angle near the loss cone angle. The observed differential number flux f [s<sup>-1</sup>m<sup>-2</sup>str<sup>-1</sup>MeV<sup>-1</sup>] was integrated over the observed energy and pitch angle range to obtain the total number flux F [s<sup>-1</sup>m<sup>-2</sup>] for each bin. Then, each R profile, representing the collision rate, was multiplied by the corresponding total number flux F to calculate the ionization rate profile for that bin. The ionization rates from all bins were finally summed to obtain the total ionization rate Q [s<sup>-1</sup>m<sup>-3</sup>] as a function of altitude. The time variation of the electron density  $N_e$  [m<sup>-3</sup>] due to production and recombination is

$$\frac{\mathrm{d}N_e}{\mathrm{d}t} = Q - \alpha N_e^2 \tag{2}$$

where  $\alpha$  [m³s<sup>-1</sup>] is an effective recombination rate proposed by Gledhill (1986). This recombination rate is a statistical estimate derived from electron density measurements during auroral events. Although image data were not always available due to bad weather conditions, electron density enhancements were clearly identified in the auroral emission altitude region. Therefore, it is reasonable to apply this value of the recombination rate in the present analysis. Since the time variation of electron density can be assumed to be negligible compared to the time resolution of the observations, i.e.,  $\frac{dN_e}{dt} \approx 0$ , the electron density was calculated as  $N_e = \sqrt{\frac{Q}{\alpha}}$ . Then, the altitude profile of electron density was calculated both with and without consideration of the effect of the mirror force. The ratio of the density with the mirror force to that without the force at 80 km altitude is used as a metric of the mirror force effect, and we refer to this as the "density ratio." We showed the density ratios at an altitude of 80 km as representative values in this paper. Finally, a comparison was made between simulations and observations of the altitude profile of electron density during the 15 seconds when the footprint of the satellite and the radar were closest.

#### 3 Results






#### 3.1 Event on December 16, 2021

#### 125 **3.1.1** Event summery

There were four events in which the ELFIN satellite and the EISCAT Tromsø radar observed simultaneously, and the electron density increased under 100 km altitude. Especially in the event on December 16, 2021, at about 07:14 UT, the satellite footprint was the closest to the radar, and they were thought to be under simultaneous observation of an electron density enhancement in the atmosphere caused by EEP. In this event, the AE index was up to 300 nT. The solar wind was quiet, with Bz at -2 nT and SYM-H at -16 nT. Figure 1 shows the footprints of the ELFIN-A for about 5 min from around 07:12 UT and the location of the EISCAT Tromsø radar. The satellite traveled southward and was closest to the radar around 7:14:30 UT. The red dots show the satellite position during the 15 seconds for which we used the observation data in this study. Figure 2a shows that the EISCAT Tromsø radar data observed electron density enhancement with a value more than  $2 \times 10^{10}$  m<sup>-3</sup> for 10 min and that it had double peaks when the satellite was the closest to the radar, one was at 104.8 km altitude, and the other was at 94.7 km altitude in the electron density profile. It is assumed that an EEP observed by the satellite made the second peak. Figure 2b shows that the electrons have a wide range of pitch angles, from small angles up to 90 degrees, and tend to be close to isotropic in the north of the radar, while most of the electrons had larger pitch angles than the loss cone angle at the southern range. It was also observed in Fig. 2c that most of the energetic electrons above 500 keV had large pitch angles, regardless of location. Figures 2d and 2e show that at the closest approach time, electrons with energy above 1000 keV were observed and that trapped electrons were the majority above 100 keV. Figure 2f shows the fraction of energy flux of precipitating, boundary, and trapped electrons separately among electrons with energy between 50 and 2500 keV. Note that the "boundary electrons" are defined as electrons observed in bins of the EPDE instrument in the satellite whose observation range includes the loss cone angle. Figure 2g is the density ratio at 80 km altitude, and Fig. 2h is the location of the ELFIN satellite. The time range in Figs. 2f and 2g is restricted between 07:13 and 07:16 because the number of electrons observed by the satellite at other times was not enough to be analyzed meaningfully. These three figures were compared in order to investigate trends in the density ratio. The density ratio decreased as the satellite went south; that is, the L-value and the invariant latitude got smaller, and simultaneously, the ratio of trapped and boundary electrons became larger. For example, the density ratio was 0.8 at maximum when the trapped electrons occupied about half of the energy flux at the location with the L-value of 10 and the invariant latitude of 72°. On the other hand, the value was 0.5 at minimum when the L-value was five, and the invariant latitude was 63°. The trapped and boundary electrons were responsible for more than 75% of the total energy flux. The density ratio was especially 0.6 at the point where the footprint of the ELFIN satellite was the closest to the radar. In other words, we can say that the density ratio became smaller as the electrons with large pitch angles were more dominant in the distributions.

Figure 1: The footprint of the ELFIN satellite (black and red dots) and the location of the EISCAT Tromsø radar (yellow star). The red ones show the satellite footprint for 15 s when both were the closest during the event.

Figure 2: Simultaneous observation data of the EISCAT Tromsø radar (a) and the ELFIN satellite (b-f), and the simulation result

(g). (a) Altitude profile of electron density observed by the EISCAT. (b) Average pitch angle distributions of the electron number flux below 500 keV were observed by the satellite. (c) Same as (b) but for that above 500 keV. (d) Average energy distributions of the electrons with the pitch angles less than the loss cone angle. (e) Same as (d) but for those more than the loss cone angle. (f) Fraction of the energy flux for precipitating, boundary, and trapped electrons. (g) The density ratio at 80 km altitude from the

simulations with/without mirror force effect. (h) Information on the geomagnetic location of the satellite: L-value, magnetic local
time, and invariant latitude. Two vertical dashed lines show the 15 s time span of the closest approach of the satellite to the EISCAT,
corresponding to the red dots in Fig. 1.

#### 3.1.2 Data of the ELFIN satellite

185

190

Figure 3 shows the pitch angle and energy distribution of the electron number flux observed by the ELFIN satellite. The time range of Fig. 3(b) includes the time of closest approach of the satellite to the EISCAT Tromsø radar. As mentioned in the introduction, the effect of the mirror force is essential mainly for electrons with large pitch angles and with energy more than 100 keV. According to Fig. 3(b), most electrons with energy less than 1000 keV had large pitch angles, though the pitch angle distribution of electrons with higher energy than 1000 keV did not show any apparent tendencies. Specifically, the number flux of 63 keV electrons, for example, was about 10<sup>6</sup> s<sup>-1</sup>cm<sup>-2</sup>str<sup>-1</sup>MeV<sup>-1</sup> with a pitch angle larger than 70° while that was less than 10<sup>5</sup> s<sup>-1</sup>cm<sup>-2</sup>str<sup>-1</sup>MeV<sup>-1</sup> with a pitch angle less than 30°. The analysis for the entire observed energy range revealed that the electrons with pitch angles almost equal to or greater than the loss cone angle account for about 72% of the total energy flux.

The distributions in Figs. 3(a) and 3(c) are the same as in Fig. 3(b) except for observation time, which has time ranges of 15 s before and after the time of Fig. 3(b). These ELFIN satellite observations suggest that the trapped particles became slilghtly more dominant for lower-energy electrons (Sergeev et al., 2012). However, the electron density measurement by the EISCAT radar consistently shows the second peak of enhancement at around 95 km altitude (see Fig. 2(a)), which corresponds to tensof-keV electrons at least according to the stopping height (Turunen et al., 2009). Therefore, although the satellite did not pass through just above the radar, it is reasonable to consider the comparison at the time of closest approach to be meaningful, since both instruments were likely observing precipitating electrons in a common energy range, and the resulting ionization profiles are comparable.

Figure 3: Pitch angle and energy distribution of the electron number flux observed by the ELFIN satellite, averaged during 15 s before (a), at the time (b), and after (c) the closest access of the satellite to the EISCAT Tromsø radar. Dashed lines indicate the loss cone angle.

#### 3.1.3 Simulation results and comparison with EISCAT observation data

Figure 4 shows three types of altitude profiles of the electron density. The black line is the EISCAT Tromsø radar data, while the red and blue lines are simulated electron density using the electron distribution shown in Fig. 3(b) as an initial condition with and without the mirror force effect, respectively. Each peak value at 87 km altitude of the red and blue lines in Fig. 4 is made by 63 keV electrons, which is the minimum value of observed energy. Because electrons with energy less than 63 keV, out of observation, must have contributed to the atmospheric electron density above 87 km altitude, we compared the simulation results to the radar data at altitudes below 85 km (see Zou et al., 2024). The density ratio between the simulations with and without the mirror force effect is 0.6 at 80 km altitude, and the value does not change significantly in the altitude range below 85 km. The altitude profile of electron density observed by the EISCAT radar at the same time is generally smaller than the values obtained from the two simulation results. The density ratio between the "with simulation" ("without simulation") and the EISCAT observation at 80 km is 1.7 (2.9), respectively. Hence, this result suggests that it might be important to consider the reduction in ionization due to the mirror force effect when studying ionization processes caused by EEP at altitudes below 85 km.

# **EISCAT** and Simulation Comparison

Figure 4: Altitude profile of electron density. Simulation results, with/without the mirror force effect, are shown by the red/blue lines, respectively. The black line shows the EISCAT Tromsø radar observation. Note that it is inappropriate to compare simulation results with the observed data above 85 km altitude because there must be an electron density enhancement due to lower energy electrons than observed by the ELFIN satellite.

#### 3.2 The results of the other events and their characteristics




We investigated four events, including the one described in Section 3. Table 1 lists the date, time, and the density ratio at 80 km altitude. The table also shows the energy flux percentage of trapped electrons, including boundary electrons (Tra. E-flux), observed by the ELFIN satellite. We calculated these values using the data of the time when the satellite was closest approach to the EISCAT Tromsø radar at the magnetic latitude of 66°.

Figure 5 shows a comparison between the EISCAT radar observations and the simulation results for all four events. The event on December 16, 2021 (the most right panel), showed that the EISCAT radar was able to derive more accurate altitude profile of electron density. For three of the events, the observed electron density agrees more closely with the simulation results with the mirror force than with those without the force, except for the event on 7 January. These results support taking the mirror force into account in calculations of electron density enhancement due to precipitating energetic electrons. On the other hand,

the EISCAT radar did not observe a clear enhancement in electron density at low altitude on October 5, 2021, as would be expected based on the conjugate simulation result.

The density ratios ranged from 0.57 to 0.80. When the value was the smallest (0.57) on December 16, 2021, trapped or boundary electrons accounted for 72% of the total energy flux. When the ratio increased to 0.80 on December 09, 2020, the energy flux carried by trapped or boundary electrons decreased to 35% of the total, indicating a significant increase in the energy flux of electrons within the loss cone. Therefore, it was found that, as mentioned in Katoh et al. (2023), electrons with large pitch angles make a difference due to the mirror force, reducing the electron density in the atmosphere resulting from the actual distribution by approximately half (see Fig. 2) compared to the traditional method, which ignored the mirror force.




# **EISCAT and Simulation Comparison** 2020-12-09 21:31 2021-01-07 19:52 2021-10-05 20:20 2021-12-16 07:14 100 95 90 Altitude [km] 85 80 75 **EISCAT** with without 70 10<sup>10</sup> $10^{11}$ 10<sup>9</sup> Electron Density [m<sup>-3</sup>]

Figure 5: Electron density comparison between the EISCAT radar observation and simulations. The black lines are the observation results, and the red and blue lines indicate the simulation results with and without the mirror force, respectively.


Table 1. Summary of conjunction events between the ELFIN satellite and EISCAT Tromsø radar

| Date       | UT    | Density ratio | Tra. E-flux |
|------------|-------|---------------|-------------|
| 2020/12/09 | 21:31 | 0.80          | 35%         |
| 2021/01/07 | 19:52 | 0.66          | 93%         |
| 2021/10/05 | 20:20 | 0.59          | 61%         |
| 2021/12/16 | 07:14 | 0.57          | 72%         |

#### 4 Discussion

To compare the overall altitude profile of electron density, the simulation results and observational data showed good agreement for the 16 December 2021 event (see section 3.2 and Fig. 5), but not clearly for the other events. One of the reason for unclear comparison results in Fig. 5 is assumed to be the low signal-to-noise ratio of the electron density profile observed by the EISCAT Tromsø radar. It is known that the signal-to-noise ratio becomes significantly low when the electron density is less than  $10^{10}$  [ $m^{-3}$ ], which has prevented accurate observation and comparison at an altitude of 80 km.

Another reason is mainly due to the spatial distance between the ELFIN satellite and the EISCAT Tromsø radar, but more accurate simulations are needed to compare its results to observational data, such as the introduction of secondary electrons. In detail, Fig. 4 also shows that the observed electron density was slightly lower than that calculated by the numerical simulation including the mirror force. This difference is most likely due to electrons scattered by whistler-mode waves (e.g., Zhang et al., 2023), as the event is characterized by intense bursty precipitations. In that case, the electron flux is expected to vary on time scales of seconds, but the satellite could not capture it within a limited latitude range conjugated to the radar.

Additionally, the close timing of the observations does not necessarily indicate that the satellite and the radar observed the same precipitation event. As shown in Fig. 6, during the closest 15 seconds, the satellite footprint (red line) includes both discrete aurora and diffuse aurora, whereas the radar only observed the diffuse aurora region. Furthermore, as seen in Fig. 3, slight changes in latitude caused some variations in pitch angle and energy distributions. Therefore, small differences in the observed precipitation event may have led to discrepancies between the simulation and the observation results.

Moreover, the difference in integration time between the two instruments (ELFIN: 15 seconds; EISCAT: 30 seconds) makes it even less certain that they observed the same precipitation. Neverthless, it is by chance that the conjunction event occurred

during an EEP event. Since events where satellites and radars can closely observe the same region are difficult to capture, it is important to utilize other incoherent scatter radars, such as the PFISR (Poker Flat Incoherent Scatter Radar) in Alaska, to find additional events and validate the mirror force effect.






Figure 6: Optical data collected by at Tromsø on 7 January 2021. The red dashed line is the satellite footprint, and the green dot indicates the radar beam point.

The ionization profile during EEP is affected not only by pitch angle distribution but also by energy distribution. Katoh et al. (2023) found that the difference due to the consideration of the mirror force becomes more evident for electrons with higher energy. Additionally, the ratio of energy flux of trapped electrons always increases with energy when EEP occurs due to the whistler-mode waves (e.g., Tsai et al., 2023). Both results support the idea that higher energy electrons are more likely to be affected by the mirror force, which reduces the electron density in the atmosphere. On the other hand, it is known that the ratio sometimes remains constant for the 50-1000 keV range during intense precipitations by high-intensity waves (Zhang et al., 2022) and that the ratio increases if precipitation is driven by EMIC waves (Capannolo et al., 2023). These studies suggest that the significance of the effect of the mirror force depends on wave strength or type, but we only found whistler-mode wave events in our conjugated observation. Therefore, it is necessary to statistically investigate the force's contribution to the electron density in the atmosphere with a focus on the type of wave.

In this study, we also examined the latitudinal distribution of the density ratio for only one event (see Fig. 2(g)). We found that the difference due to the mirror force depends on latitude and that the ratio is more likely to be smaller as latitude decreases, even while electrons seemed to be precipitating from the radiation belt. This is because the distribution of electrons depends on latitude. Specifically, this is because the percentage of trapped electrons was more significant at lower latitudes. It is also statistically consistent with Qin's finding that the smaller the L value, the larger the ratio of energy flux of trapped electrons. However, the effects of the mirror force, i.e., the variations in density ratios due to distribution changes in both the energy and pitch angle, are not yet fully understood, so more ELFIN satellite observation events should be used to examine the latitudinal distribution of the ratio.

Finally, this study focused on the magnetic mirror force and included its effects in the simulation to see how it changes the electron density enhancement in the atmosphere. It was mentioned in Katoh et al. (2023) that some electrons move back away from the atmosphere, i.e., go up again. However, even though the ELFIN satellite observed the flux of upgoing electrons, a comparison between the observational data and the simulation results has yet to be made. In the future, it can be verified whether the upgoing electrons observed would be consistent with the simulation results by improving the simulation code so that the energy and pitch angle distributions of the reflected electrons can be calculated.

#### Conclusion

This study aims to evaluate the effect of the mirror force on precipitating electrons and, consequently, atmospheric electron density by focusing on the pitch angle of electrons. We used ELFIN satellite data and a simulation developed by Katoh et al. (2023). We found that the electron density due to EEP can be about 40% smaller when the effect of the mirror force is considered than when it ignored the effect as in previous studies (e.g., Murase et al., 2023). The simulation results were compared with simultaneous data from the EISCAT Tromsø radar. For an event where the altitude profile of electron density was accurately determined from the EISCAT radar on December 16, 2021, the simulated electron density profile, including the mirror force effect, is closer to the actual density profile from EISCAT than when ignoring the effect. Furthermore, although this is not a conjunction event between the ELFIN satellite and the radar, some pitch angle distribution made the ratio reach 50% at maximum. In other words, these results suggest the importance of considering the pitch angle distribution of electrons and the effect of EEP on the simulation of electron density enhancement in the atmosphere, or the result would differ by about 50%. Additionally, in order to accurately estimate the effects of atmospheric ionisation, it is desirable for electron instruments with Low Earth Orbit (LEO) satellites to observe pitch angle distributions with high resolution, which can distinguish between trapped and precipitating particles.

## **Author contributions**

Y. O. designed this study. Y. K. instructed T. T. on the use of simulation he developed. A. A., V. A., and X. Z., collected and proceeded the data observed by the ELFIN satellites. Y. O., Y. K., A. K., Y. T., and M. F. supervised the work. M. F. and T. T. analysed the data. T. T. made the figures and drafted the manuscript. All authors discussed the results and approaved the manuscript.

### Acknowledgments

This research was supported by the Grants-in-Aid for Scientific Research (20K04052, 20KK0313, and 23H05429) by the Ministry of Education, Science, Sports and Culture, Japan.

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
