# Peer review of "Effects of geomagnetic mirror force and pitch angles of precipitating electrons on ionization of the polar upper atmosphere"

_EGUsphere, 2025_

## Author Response (AR1)

Referee comments and replies (RC1)

**RC1-1**

- •Why use the auroral electrojet index in place of Kp or a more robust index for this energy range of electrons? The AE index is primarily an index for aurora as it measures ionospheric currents in the auroral oval. Additionally, as you mention, they are sourced from the radiation belts where the precipitation mechanisms are often different from more general auroral electrons. It would be more useful to choose time periods with varying Kp or maybe even DST instead of AE for this study.
- → The reply to RC1-2 also addresses the contents pointed in RC1-1.

**RC1-2**

- •All five of the events that are discussed take place in mostly undisturbed times. It is mentioned that the density ratio does not appear to trend with AE, but I disagree with this because none of the events that are shown are during geomagnetically disturbed conditions. Even the event with 350 AE shows virtually no Kp enhancement indicating very little geomagnetic activity. I do not think you can claim the density ratio does not depend on activity (in the text you claim AE) when the events chosen are not very different in terms of enhancements. I would reword the conclusion to state that these times represent mostly undisturbed time periods.
- → As the referee pointed out, most of the simultaneous observation events obtained in this study occurred during undisturbed periods. The purpose of this study is to investigate the effect of the mirror force, regardless of the index values. Then, it is true that the AE index is not necessarily an index for finding EEP events. Moreover, events do not always have to be associated with a specific index in this study. Therefore, we deleted some sentences including the AE index.

**RC1-3**

- •I would suggest showing an additional panel with four other plots of the density profiles for the other four events instead of just for one event. I think the most compelling is the plot that shows the simulation that includes the mirror force agrees better with EISCAT than the one without.
- → A new panel (Fig. 5) and some relevant results, including the four events, have been added. One event was removed because it did not satisfy the conjunction condition.

**RC1-4**

- On that same note, are there any other events using EISCAT and ELFIN that are during more geomagnetically active time periods? If not have you considered adding additional radars such as HARRP or the Canadian ISRs? I think having a range of density profiles based on geomagnetic activity would be compelling. It could also be a very good follow-on paper as well.
- → There were no other events using EISCAT and ELFIN during high geomagnetically active periods. Utilizing other ISRs is added in the text as one of the future works.

Specific Comments: (RC1)

**We have revised the sentences according to all comments from 1 to 7.**

\_\_\_\_\_

- 1. Line 27: The first sentence doesn't make grammatical sense. Consider revising.
- 2. Line 32: There are other references that should be added here that discuss chemical changes from EEP. Some to consider are: ...
- 3. Line 35: "magnetic field" not "magnet field"
- 4. Line 47: "electrons" should be "electron"
- 5. Line 50: Remove "between"
- 6. Line 70: It may be worth adding the equation here on how you defined the loss cone mathematically.
- 7. Line 79: Grammar. "data of an electron" doesn't make sense. Please revise.

- 8. Line 119-121: I'm confused about Figure 2b. The sentence claims that the < 500 keV electrons have larger pitch angles north of the radar than south. However, Figure 2b shows the opposite. I see more electrons north of the radar (according to Figure 1 before crossing the EISCAT instrument) with pitch angles below the loss cone angle (preassembly the black line in the plot). This tells me that more electrons are precipitating north of the radar than south according to the figure. Is the figure incorrect or is the explanation incorrect?
- → We have revised the sentence as it was confusing. The figure is correct.

9. Line 123: change 'major' to 'the majority'

**→ We have revised it.**

10. Line 210: I think I might understand this sentence, but it should be changed to be clearer.

"For instance, precipitating electrons from the radiation belt due to the waveparticle interaction are thought to have a relatively small pre/tra ratio because they originally had a large pitch angle to maintain the region"

Are you saying that the large pitch angles for radiation belt electrons required for the radiation belts to exist? Because otherwise, if all the electrons were at low pitch angles, would they precipitate? I find the wording of this confusing.

→We agree that the wording is confusing. What we meant to convey is that electrons in the radiation belt need to remain outside of the loss cone to stay trapped. We have removed a paragraph including the sentence because the AE index and the aurora type are not the focus of this study.

**Referee comments and replies (RC2)**

**RC2-1**

- •Descriptions of the instruments and data should be clarified. Section 2.1 first mentions "the ELFIN satellite" but then continues to describe a pair of satellites. The loss cone angle, which is critical for the study, is shortly explained, but exact formulas and the magnetic field model used in the calculations are not included and should be added. Three EISCAT radars are mentioned, but it remains unclear that which radar was actually used, where the radar beam was pointed, and how the data were analysed. In particular, radar power calibration is not mentioned at all, although it is critical to get reliable electron densities. Radar operation mode and how the data were analysed and calibrated should be explained.
- → The ELFIN satellites consists of two CubeSats, but only one of them was in conjugate observation. This study only used data from the satellite in conjugate observation. Section 2.1 has been revised with taking account for the comment.
- → We identified that only the EISCAT Tromsø UHF radar (operated with the beata pulse code) was used in this study. The relevant descriptions, including the radar pointing direction and data analysis procedure, have been revised accordingly in section 2.2.

**RC2-1**

•Description of the simulation should be expanded and clarified. The way the electron production rates are calculated should be shortly outlined. There are references to three different versions of the MSIS model, but no information about which version was actually used. This should be clarified. The exact inputs to the model remain unclear (there is a list that begins with the words "such as", giving the impression that the list may not be complete). This should be replaced with a complete list of the inputs. Also, description about the way the Lorentz force is modelled is unclear and very short. This part

of the modelling that is critical for the paper should be explained in sufficient detail.

- → We identified that only one MSIS-model (NRLMSIS 2.1) was used in pyMSIS, so we have revised the explanation and citation.
- → The information on the simulation model used in this study is described in detail in Katoh et al. 2023, so we have revised the citation accordingly.

**Rc2-3**

- •Comparison with EISCAT radar data is eventually made for only one out of five events. It is then mentioned that the simulation results and radar measurements were in good agreement in one event, "but not necessarily for the other events". It is very unclear what "not necessarily" means here, and why conjunctions with the EISCAT radar were initially chosen but the data are not shown. I recommend that the authors should present electron densities measured by EISCAT for all the events and then discuss reasons for possible differences.
- → We agree with the reviewer 's comment and modified the expression. Of the multiple simultaneous observation events, we have described in detail one event in this paper where t the altitude distribution of electron density measured by EISCAT was clearly identified. According to the reviewer 's comment we added a new panel (Fig. 5) and some relevant results, including the four events. One event has been removed because it did not satisfy the conjunction condition.

**RC2-4**

- •The manuscript would benefit from a thorough language check.
- $\rightarrow$  We, especially corresponding author, have revised the manuscript as much as possible in response to the reviewer's comment.

**Specific comments (2人目)**

**We have revised the sentences according to all comments from 1 to 18.**

- 1. Lines 15-18:
- EISCAT radars are mentioned here, but the 40% difference is only between two simulation results. EISCAT should be mentioned only later when discussing the actual comparison with the radar data.
- 2. Line 27: "It has been focused on the impact..."
- I cannot understand this sentence.
- 3. Line 30: "...electron density enhancement at about 50 km altitude..."
- Please clarify the energy range discussed here. > 50 keV is mentioned in line 27, but electrons that reach 50 km altitude must be much more energetic.
- 4. Line 39: "...altitude change of electron using..."
- I cannot follow this. Should the text say, "altitude change of electron flux", or something similar?
- 5. Lines 40-41: "Recently, ... (Lehtinen, 1999)"
- The reference is 26 years old; I would not call it recent.
- 6. Lines 50-51: "However, the satellite, ..."
- Which satellite? Is this still about Reimei?
- 7. Line 54: "...the satellite..."
- which satellite? Or satellite data in general?
- 8. Line 64: "...two CubeSats ..."
- The section title says, "ELFIN satellite", but apparently this is a pair of satellites?
- 9. Lines 72-74
- The description of EISCAT radars is extremely short. Please add information about which radars were actually used, and how were the data analysis and power calibration conducted. The power calibration is critical here, because

any inconsistency in the calibration will bias the electron densities that are used in the comparisons.

- 10. Lines 77-78: "(Lucas, 2022; Emmert et al., 2020; Emmert et al., 2022; Picone et al., 2002; Celestrak; Matzka et al., 2021)."
- The references point to three different version of the MSIS model (NRLMSISE-00, MSIS 2.0, and MSIS 2.1), as well as to CelesTrack that i cannot immediately connect to this topic. Please refer only to the relevant model.
- 11. Lines 81-82: "considering the force, the magnetic field at a distance of Larmor radius from the magnetic field line leading to Tromsø was given to satisfy  $\nabla \cdot \mathbf{B} = 0$ "
- Does "the force" refer to the Lorentz force? Also, I cannot follow the rest of the sentence. Please clarify how the Lorentz force was modelled.
- 12. Line 88: "...the ELFIN satellite as the initial condition of simulations to calculate the altitude profile of collision rate..."
- Are the inputs energy fluxes in energy & pitch angle bins?
- "Collision rate" is a bit vague term here, since a collision does not necessarily lead to ionization. I would rather call this "ion production rate".

**13. Line 89:**

- Please provide a short description about how R is calculated.
- 14. Line 92: "...vary significantly depending on the initial pitch angle."
- Does this refer to the initial pitch angle distribution?
- 15. Lines 94-95: "The differential number flux...was integrated according to the observed range of energy and pitch angle to obtain the total number flux F... It was multiplied by R to obtain the ionization rate Q..."
- From lines 87 to 89 I understood that R is calculated for each energy and pitch angle bin separately. If this is the case, then the latter description cannot be correct. Were the R values in each bin perhaps multiplied with the corresponding fluxes, and the products were integrated?

**16. Line 99:**

- Please provide a short description about how  $\alpha$  is calculated.

- 17. Line 102: "the effect" -> "the mirror force"
- 18. Subsections in Section 3.
- Subsections 3.1, 3.2, and 3.3 are all about the same event, while 3.5 is about the other events. The current subsections 3.1, 3.2, and 3.3 should be either merged under a common subsection (to become 3.1.1, 3.1.2, 3.1.3), or the titles of 3.1, 3.2, and 3.3 should be such that it becomes clear to the reader that these sections are only about one single event.

- 19. Line 111-112: "..., and geomagnetic pulsations were found."
- What kind of pulsations and what do they tell us? If the pulsations are discussed it would be nice to have the data included in Figure 2.
- $\rightarrow$  The geomagnetic pulsation is not a key point in this study, so we have removed the sentence.
- 20. Lines 118-119: "It is assumed that an EEP observed by the satellite made the second peak"
- Do the authors have a suggestion about what caused the upper peak? If it is less energetic electron precipitation, are there two particle populations with different peak energies at the same field line simultaneously?
- → Such two particle populations would be possible (e.g., Figure 1 of Nishiyama et al., 2011), but it is hard to clarify the distribution only from the ELFIN data. The point is EEP made the electron density enhanced under 100 km altitude, not upper one.
- 21. Lines 125: "...bins of equipment..."
- Does "equipment" here refer to the EPED instrument in the satellite?
- $\rightarrow$  Yes, it does. We have added its explanation in the text.
- 22. Lines 159-160 and Figure 3: "..., which has time ranges of 10 s before and after the time of Fig. 3(b)."
- The figure shows three 15 s time intervals with 5 s overlap between subsequent intervals. Since the idea is to demonstrate that the precipitation

did not change with latitude, it may be better to show non-overlapping intervals.

- → We can agree with it. However, we believe that overlapping time intervals are the most appropriate way to present the data. One reason is that 15 seconds are needed to obtain reliable flux averages for almost all energy bins, especially because the number flux of electrons above 1000 keV is small. On the other hand, using a non-overlapping 45-second window would be too long for comparison, since the satellite moves rapidly through the region at an altitude of approximately 400 km.
- 23. Lines 171-172: "Each peak value at 87 km altitude of red and blue lines in Fig. 4 is made by 63 keV electrons, which is the minimum value of observed energy"
- If the peaks at 87 km are artifacts and everything above 87 km is biased because the satellite does not observe < 63 keV electrons, why is the upper part shown at all?
- → The region above 87 km altitude is included in the figure to show the actual peak altitude of electron density. The comparison between EISCAT and simulation results is made for the region below 85 km altitude.

We have revised the sentences according to all comments from 24 to the end.

- 24. Lines 191-192: "...or boundary electrons had 35% energy flux, which is less than half of the former; thus, the pitch angle distribution was close to isotropic."
- The fraction of trapped or boundary electrons that corresponds to an isotropic distribution depends on the loss cone angle. What is this fraction and how large is the loss cone angle in this case?
- 25. Lines 192-193: "In addition, it was ~10 min before the maxima of the horizontal geomagnetic component in Tromsø, and the peak of the AL

decreased."

- Please clarify this sentence.
- 26. Lines 197-198: "Focusing on whole events, the MLT distribution shows that differences due to the mirror force appear in the order of before midnight, at night, and in the morning. On the other hand, the AE index does not look like the key to the difference"
- This is a very far-reaching conclusion from only five data points. The data do not exclude the described distribution, but do not confirm it either.
- 27. Line 203: ""pre/tra ratio""
- I cannot find the column "pre/tra ratio" from Table 1.
- 28. Lines 208-209: "Although the images could not be analyzed in this study because they were unavailable, ..."
- What images? I would guess that this refers to ground-based optical data, but no images are mentioned anywhere else in the manuscript.
- 29. Lines 211-212: "...for the 7 Jan. event when the diffuse aurora was observed."
- There is no evidence of diffuse aurora given in this manuscript. How did the authors conclude that this was a diffuse aurora event?
- 30. Lines 215-216: "...the simulation results and observational data were in good agreement for the December 16 event (see Fig. 4), but not necessarily for the other events"
- This remains unclear for the reader, because radar data for the other events is not shown. Please add EISCAT Ne at 85 km altitude in Table 1 to enable this discussion.
- 31. Lines 218-219: "Fig. 4 also showed a minor disagreement between observation and simulation results."
- What exactly is this minor disagreement? The overall larger Ne in the simulation, some other detail in the results, or both?